# MoDA: Mixture of Domain Adapters for Parameter-efficient Generalizable Person Re-Identification

## Abstract

Domain Generalizable ReID task has garnered much attention in recent years, as a more challenging task but more closely aligned with practical applications. Mixture-of-experts (MoE) based methods has been studied for DG ReID to exploit the discrepancies and inherent correlations between diverse domains. However, most of DG ReID methods, including MoE-based methods, have to full fine-tune the large amount of parameters of backbones, classifier heads and experts. And in the set of DG ReID, the number of person IDs is particularly large which results in that parameters of classifier heads increases sharply. And make it difficult for MoE-based method to scale up to larger vision models. For this motivation, we propose a novel MoE-based DG ReID method, named mixture of domain adapters (MoDA), to mitigate the issues mentioned above. We apply Adapter-tuning and CLIP to DG ReID in a parameter-efficient way. Extensive experiments verify that MoDA achieves competitive end even better results with state-of-the-art methods with much fewer tunable parameters.

## 1 Introduction

Person re-identification (ReID) has emerged as a pivotal research area in computer vision. Despite the existence of numerous studies in this task, it is still attracting significant attention and garnering considerable importance. There have been many works that can get great performance on ReID benchmarks in a conventional scenario. Based on the differences of backbone networks, these methods can be roughly classified into two categories: CNN-based and ViT-based. However, when these methods are confronted with a completely unseen domain, the performance drops significantly. This phenomenon is commonly attributed to domain shift and domain conflict. To tackle this problem, some domain adaptation (DA) and domain generalization (DG) ReID methods are proposed. DA methods usually can access a part of target domain data. And then they try to adapt the model already trained with source domain data to the target domain. When it comes to domain generalization, the task becomes even more difficult. The model can only use images from source domains to optimize and is not allowed to access any target domain image during the training time. By means of such a task definition, to enhance model's generalization ability and robustness when facing out-of-distribution data.

Given these reasons above, the DG ReID task has garnered much attention in recent years, as a more challenging task but more closely aligned with practical applications. Convential ReID methods usually train and test on the same domain. However, in the context of DG ReID, the training and testing of models are performed on diverse domains, which are respectively referred to as the source domain and target domain. Many prior DG ReID methods Song et al. (2019) Zhao et al. (2020) only utilize one individual model and train the model on a hybrid dataset consists of all samples from different source domains. And the model is directly used to test on unseen target domains. These methods achieve good performance by extracting domain-invariant features. However, they disregard the discrepancies and inherent correlations between diverse domains, which may provide more discriminative and complementary information to help generalize better. For this reason, mixture-of-experts (MoE) based methods has been studied for DG ReID. A common practice of MoE-based methods is to train domain-specific expert network on each source domain. Then they integrate multiple experts by calculating the relevance of the test sample and source domains to get one aggregated

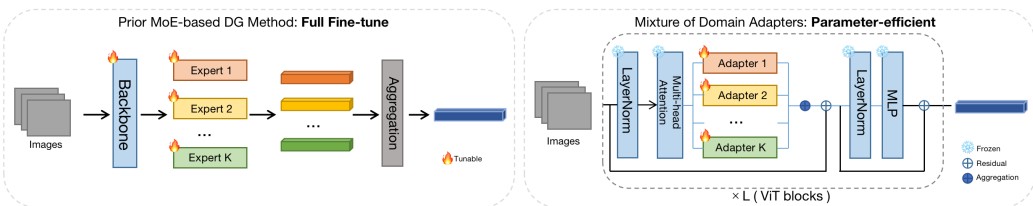

Figure 1: Difference between prior MoE-based DG ReID methods and our more parameter-efficient method. Left shows that prior MoE-based DG method need to optimize the whole model including the backbone and experts, parameters of which usually increase linearly with the increasing of the number of person IDs. And right shows that in our method, only adapters need to be optimized and the number of the parameters is not related with the person IDs.

feature. The existing MoE-based DG ReID methods get better performance but have the following issue: the number of model's parameters scales linearly with the number of source domains due to the increase of the number of experts as shown in Figure 1. This makes it difficult for MoE-based methods to scale up to larger vision models such as Vision Transformer (ViT). Although there are some methods try to minimize the number of expert parameters, there are still a large amount of trainable parameters need to be updated. This is because the backbone and classifier heads contribute the most parameters in actual. Especially under the DG ReID setting where the number of person IDs is particularly large, the number of parameters of linear classifier heads significantly linearly increases. While, these MoE-based methods always need many classifier heads, at least one for each source domain, to optimize the model.

For this motivation, we propose our method, named mixture of domain adapters (MoDA), to mitigate the issues mentioned above. Adapter Houlsby et al. (2019) and the most popular vision-language pretrained model CLIP Radford et al. (2021) are applied in our method. Adapter is a bottleneck module used for parameter-efficient fine-tuning (PEFT) and will be used as domain-specific experts in MoDA. CLIP is a powerful pretrained model which aligns texts and images in one feature space. Moreover, the contrastive losses that CLIP uses can provide similar discriminitive guidance of optimization to cross-entropy loss, which means it can substitute the ID loss of ReID to reduce the classifier heads' parameters partly. Some works Vidit et al. (2023) with the image encoder of CLIP have already shown good performance in other computer vision DG tasks. As Figure2 shows, different from CNN-based methods, the adapters are inserted in each block. Thus, the backbone+expert heads structure of prior MoE methods is not compatible for MoDA. Due to this reason, we propose a blcok-aware voting network to make MoE can be used for this kind of model which consists of blocks. It enables the model integrate expert adapters in a more fine-grained way by generating aggregation weight for each block, shown in Figure 2(d).

In summary, we make the following contributions in our work:

- To the best of our knowledge, our work is the first one to exploit CLIP to DG Person ReID and also the first one to attempt PEFT methods ,such as adapter, for DG Person ReID.
- We propose a novel DG ReID framework, called Mixture of Domain Adapters (MoDA), which achieves parameter efficiency while scaling up MoE-based DG Person ReID methods to a larger vision model. And benefiting from the block-aware design, MoDA can also help model to mix experts in a more fine-grained way.
- Extensive experiments demonstrate that MoDA has achieved competitive and even better performances with existing methods in a more parameter-efficient way.

## 2    RELATED WORKS

**Domain Generalizable Person Re-Identification.**  Most of existing ReID methods suffer from significant performance degradation when they encounter the problem of domain generalization (DG). This indicates that the model needs to possess a high generalization ability since it cannot access any data of target domains. To tackle the domain conflict problem we mentioned above,

many DG ReID moethods are proposed. DG ReID methods are required to be trained on accessible source domains and then tested on unseen target domains. The most prior approaches Zhao et al. (2020) Song et al. (2019) Choi et al. (2020) Jin et al. (2020) are to combine all the source domains together as a hybrid dataset, then they train the model with the hybrid dataset and directly test on the target domain. These methods aim to learn a common feature space which works for all diverse domains. They commonly employ: 1) Disentanglement learning Zhang et al. (2021). 2) Meta-learning Zhao et al. (2020), which is utilized to simulate the domain generalization scenario where no target domain data is accessible during test period. They learn domain-invariant features but tend to overlook domain-specific features and the correlations among them. 3) Normalization-based methods Jin et al. (2020) Choi et al. (2020), which is to investigate the statistical distribution discrepancies across different domains, for instance, by minimizing the domain gap through domain alignment. However, these methods may ignore the discriminative characteristics of each domain and the interrelations between them.

To address this issue above, the mixture of experts(MoE) paradigm has been introduced to DG Person ReID Dai et al. (2021). The proposed method RaMoE intergrates every domain specific expert's feature as one single aggregated feature according to the target domain's inherent relevance w.r.t. source domains. Building upon RaMoE, META Xu et al. (2021) integrates normalization-based techniques by leveraging Instance Normalization (IN) and Batch Normalization (BN) to address the linear scaling issue of MoE with increasing source domains. It also incorporates a global branch to combine domain-specific and domain-invariant representations. ACL Zhang et al. (2022) also integrates the design fashion of MoE into their method. However, the existing methods need to full fine-tune the whole models with a large amount of parameters including the backbones and experts. Our method aims to mitigate this issue.

**Vision-Language Pre-trained Model.** Recently, more and more studies on vision-language pre-training(VLP) has shown that VLP can significantly improve the performance of many vision tasks by aligning image features and text features in one common feature space with large-scale multi-modal data. CLIP Radford et al. (2021) and ALIGN Jia et al. (2021) are good practices, which utilize both a image encoder and a text encoder. And they use contrastive loss InfoNCE to train on datasets consists of image-text pairs. Many researchers have started applying CLIP to various downstream tasks. However, the labels of ReID tasks are indexes, and there are no specific words to describe the images. Thus, it is hard to adopt the VLP models in ReID in a common way. And CLIP-ReID Li et al. (2022), which is inspired by prompt-tuning method CoOp Zhou et al. (2021), is the first study to apply CLIP for ReID tasks.

**Parameter Efficient Fine-tuning.** Conventional approaches usually fine-tune all the parameters (full fine-tuning) of pre-trained models. Due to the significantly increasing overhead of full fine-tuning VLP models, more and more parameter-efficient fine-tuning (PEFT) methods are proposed. These methods only fine-tune a small number of parameters and freeze most of them. They aim to reduce model computation and the number of tunable parameters as much as possible without sacrificing too much performance or even outperform full fine-tuning. Many methods have been widely applied in NLP tasks for large language models. Prefix-tuningLi & Liang (2021) and prompt-tuningLester et al. (2021) utilize soft prompts by appending trainable prefix tokens to the first layer or every layer, and only fine-tunes these additional tokens during training. LoRAHu et al. (2021) tries to learn low-rank matrices to approximate parameter updates. Houlsby et al. (2019) inserts small bottleneck modules named adapters to each layer of the pretrained model and only fine-tune the adapters during training. Recently, parameter-efficient methods are also studied for computer vision tasks. Jia et al. (2022) proposes visual prompt tuning for large-scale Transformer models in vision in a similar to prompt tuning. Lin et al. (2022) adapts a pretrained CLIP image encoder for action recognition tasks with low computation. And Yang et al. (2023) uses adapters to adapt more image models to video tasks without full fine-tuning.

Notably, although there are already some researchers that also have studied how to apply MoE on models with adapters, such as AdaMix Wang et al. (2022) and MixDA Diao et al. (2023), but they use a stochastically routing way or simply generate weights with MLP to choose and integrate adapters. While we use a block-aware voting network by calculating the relevance. AdaMix is used for common NLP tasks and MixDA tries to inject domain knowledge into language models rather than studying on domain generalization. However, our work is specifically for DG vision task.

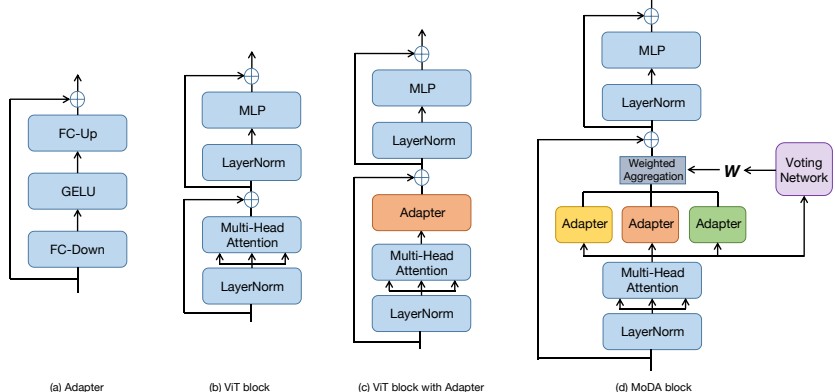

Figure 2: We show the architecture of the Adapter, ViT block with Adapter and our method MoDA block. The standard achitectures of Adapter and ViT block are shown within (a) and (b), respectively. And in our proposed MoDA block, we set Adapters after the multi-head attention layer to implement an MoE architecture as (c) shows. An additional block-aware voting network is set to mix the Adapters. Only the parameters of adapters and the voting network need to be updated during training.

## 3 METHOD

### 3.1 OVERVIEW AND PRELIMINARY

In our work, we adopt a similar two-stages training approach as CLIP-ReID Li et al. (2022) as shown in Figure 3. In the first training stage, to fully exploit the cross-modal description ability in CLIP, we utilize both text encoder $\mathcal{T}$ and image encoder $\mathcal{I}$ of CLIP. Similar to CLIP-ReID, we assign ID-specific tokens to each person. Specifically, the text descriptions fed into $\mathcal{T}$ are designed as "A photo of a $[X]_1[X]_2[X]_3...[X]_M$ person", where each $[X]_m$ is a learnable text token with the same dimension as word embedding. By optimization of two contrastive losses inspired by CLIP (image-to-text loss $\mathcal{L}_{i2t}$ and modified text-to-image loss$\mathcal{L}_{t2i}$), the trained ID-specific tokens can provide discriminative information of each ID. And these tokens will be treated as domain prototypes for the second stage to compute the relevance w.r.t different source domains. Notably, only tokens $[X]_m$ are optimized while the encoders are frozen. The loss $\mathcal{L}_{i2t}$ is formulated as, specifically, $i$ denotes the index of the images within a batch with a batch-size B:

$$\mathcal{L}_{i2t} = -log\frac{exp(\langle V_i, T_i\rangle)}{\sum_{a=1}^{B} exp(\langle V_i, T_a\rangle)} \tag{1}$$

where $V_i$ and $T_i$ are image feature and text feature produced by $\mathcal{I}$ and $\mathcal{T}$, respectively. And $\langle V_i, T_i\rangle$ represents for inner product to compute similarities. For $\mathcal{L}_{t2i}$, different images in a batch probably belong to the same person, so one ID-specific token may have more than one positive image samples. Therefore, text-to-image loss is modified to:

$$\mathcal{L}_{t2i}(y_i) = \frac{-1}{|P(y_i)|} \sum_{p\in P(y_i)} log\frac{exp\langle V_p, T_{y_i}\rangle}{\sum_{a=1}^{B} exp(\langle V_a, T_{y_i}\rangle)} \tag{2}$$

where $P(y_i)$ is the set of indices of all positives for $T_{y_i}$ in the batch.

In the second stage, we propose an MoE model with expert adapters and global adapters, which is to extract domain-specific features and domain-invariant features, respectively. During training period, we can only access the datasets from source domains to train the DG model. And the optimized DG model will be test on unseen target domain directly. We assume there are $K$ source domains, denoted as $D = \{D_k\}_{k=1}^{K}$. And for the $k$-th domain, there are $M_k$ Identities and $N_k$ images. Different from CLIP-ReID, we adopt adapter Houlsby et al. (2019) due to its simplicity and parameter-efficiency rather than full fine-tuning the ViT. And this also concurrently alleviates the optimization challenge posed by ViT's abundant parameters as well as the scalability limitation of MoE-based approaches on larger models. In addition, adapters also help prevent catastrophic forgetting from insufficient full fine-tuning which can destroy the generalization of foundation models like CLIP. As Figure

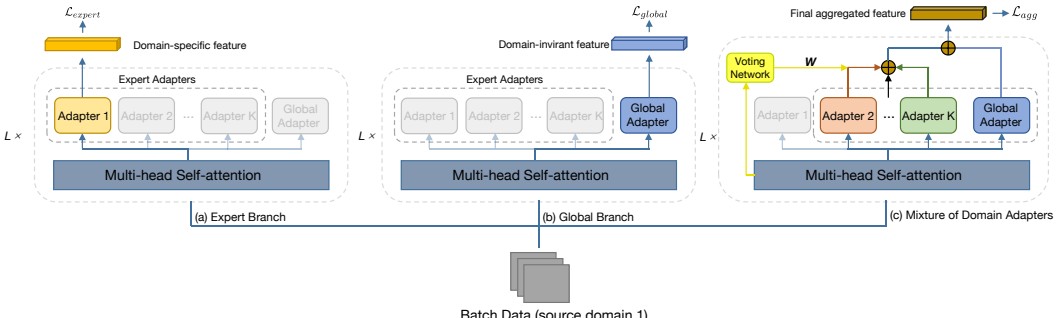

Figure 3: The framework of Mixture of Domain Adapters(MoDA). For each batch, we randomly pick one domain to simulate the unseen target domain. And the batch data will be send to 3 branches to compute responding losses, respectively. As shown in the figure, we choose Domain 1 as the mimic target domain. The expter branch and global branch is used to optimize the expert adapter and the global adapter. And the branch(c) is used for the voting network. And in the whole procedure, only the parameters of the expert adapter, the global adapter and the voting network are tunable. While all other parameters are frozen.

2(a) shows, adapter is a simple bottleneck module with two FC layers and GELU. The subsequent sections extensively delineates the second stage.

## 3.2 DOMAIN-SPECIFIC ADAPTERS

As well as the prior MoE-based methods for generalizable ReID, our method assigns an individual adapter as the domain expert of each source domain, denoted as $A_k$ standing for the domain-specific adapter of the $k$-th domain. We aim for each expert to capture characteristics features of its corresponding source domain, which are unique but can provide complementary information different from other domains. Inspired by META Xu et al. (2021), we similarly utilize a global adapter as the global branch to extract domain-invariant features.

In every ViT block, we set K adapters after the multi-head attention layer as Figure 3 shows.

**Expert Adapter.** For a $k$-th domain's image from the $n$-th ID, we denote it as $x_n^k$. As Figure 3(a) shows, for each $x_n^k$, we let it go through ViT blocks with only the corresponding domain adapter $A^k$ to obtain domain-specific feature $F_x^k$. The computation of each expert block can be written as:

$$f_l^{k'} = f_{l-1} + A^k(MA(LN_1(f_{l-1}))) \tag{3a}$$

$$f_l^k = f_l^{k'} + MLP(LN_2(f_l^{k'})) \tag{3b}$$

where $f_{l-1}$ and $f_l$ denote the input and output of the $l$-th ViT block. And only the [CLS] token, rather than all tokens, of the output of the last layer is treated as the feature $F_x^k$.

Similar to CLIP-ReID, we use the ID-specific text features obtained in the first training stage to calculate the image-to-text cross-entropy $\mathcal{L}_{i2tce}$ with label smoothing:

$$\mathcal{L}_{i2tce}(i) = \sum_{k=1}^{M^k} -q_k log \frac{exp(\langle V_i, T_k \rangle)}{\sum_{a=1}^{M^k} exp(\langle V_i, T_a \rangle)} \tag{4}$$

where $T$ are just the ID-specific tokens in the corresponding $k$-th domain of the current image. Notably, we find that with adapters, the usage of ID loss $\mathcal{L}_{id}$ has little influence on model performance. We speculate the reason might be that the image-to-text loss $\mathcal{L}_{i2t}$ and ID loss $\mathcal{L}_{id}$ proposed by CLIP are inherently both cross-entropy losses, thus can provide approximate constraints and optimization capabilities. However, the additional classifier head leads to substantially more tunable parameters under the DG ReID experiments' configuration. Therefore, to additionally enhance the parameter-efficiency of our model, we discard the ID loss which is widely used in ReID tasks.

Triplet loss is still employed for expert adapter in our methodology:

$$\mathcal{L}_{triplet} = max(d_p - d_n + \alpha, 0) \tag{5}$$

And, the loss for expert adapter is formulated as:

$$\mathcal{L}_{expert} = \mathcal{L}_{i2tce}^{e} + \mathcal{L}_{triplet}^{e} \tag{6}$$

where the superscript $e$ which stands for expert is to distinguish from the losses for global adapter that will be mentioned below.

**Global Adapter.** Except for the domain-specific experts, we also maintain an individual global adapter to learn a common feature space to extract domain-invariant representations. The extracted features are invariant to source domain shifts and more robust compared to any individual domain adapter. As Figure3 (b) shows, for each sample $x$ from any domain, we pass it through the global adapter to obtain global feature $F_x^g$. The computation of each global adapter block is actually the same as the expert adapter block's. But we will combine all text features from different domains together to perform $\mathcal{L}_{i2tce}$. And, the loss for global adapter is formulated as:

$$\mathcal{L}_{global} = \mathcal{L}_{i2tce}^{g} + \mathcal{L}_{triplet}^{g} \tag{7}$$

Notably, during the training process, the parameters of the ViT backbone are frozen and shared across different adapters, only the adapters' parameters need to be updated.

### 3.3 MIXTURE OF DOMAIN ADAPTERS

Now, we have both domain-specific and domain-invariant features, but the target domain is inaccessible during DG ReID training. To improve generalization on unseen domains, we propose a novel MoE-based approach specially for ViT with adapters to integrate domain-specific features and get aggregated representations which are more robust.

In most existing MoE-based methods, they learn diverse experts for different source domains and then calculate the relevance between the test image which is from unseen target domains and source domains. Subsequently, as guided by the relevance, multiple source domain features are mixed to create a new aggregated feature. In prior CNN-based approaches, the standard voting network needs to generate query features from backbone feature maps and calculate relevance with prototypes of different source domains. The domain's class centers are usually used as its prototypes. However, this is not compatible for ViT-based models that are composed of blocks. Different from CNN's backbone + expert heads structure, in ViT-based methods, to obtain the final features from k experts, we have to forward the whole ViT model $k$ times. This will sharply increase the computation for each sample. To mitigate this issue, we propose Mixture of Domain Adapter (MoDA) and a block-aware voting network to enable the model to mix features from multiple domain-specific experts at the block level. The kind of block-wise approach, compared to the sample-wise approach, also makes the model conduct a more fine-grained feature mixing in each block rather than using one single weight for all blocks.

As mentioned before, in the first stage, we completely follow CLIP-ReID to generate ID-specific tokens for each identity. These tokens will be treated as domain-specific prototypes $P = \{P_k\}_{k=1}^{K}$ to describe the characteristics of different source domains.

In the second stage, we adpot episode learning algorithm following Xu et al. (2021) to simulate the testing scenario where target domains are unseen. For a sample from the $k$-th domain, we denote it as $x^k$. We assume $x^k$ is the current training sample input to the model. We firstly let it pass to blocks with the $k$-th domain-specific adapters $A_k$, then compute the expert loss $\mathcal{L}_{expert}$. And in this iteration, the $k$-th domain will be seemed as the unseen target domain, and the remaining $K-1$ domains $\{D_i\}_{i=1,i\neq k}^{K}$ will be seemed as source domains $\{D_i^s\}_{i=1}^{K-1}$. Then, $x^k$ is passed to the MoDA blocks only with the remaining $K-1$ expert adapters $\{A_i\}_{i=1,i\neq k}^{K}$, as shwon in Figure3(c).

For the $l$-th MoDA block, we denote the input and output as $z_{l-1}$ and $z_l$. In each MoDA block, $x^k$ will be fed into LayerNorm and Multi-head attention(MA) first. And we pass the output of MA $z_l'$ to all $K-1$ expert adapters concurrently. Then we can get can $K-1$ diverse intermediate features $\{h_l^i\}_{i=1,i\neq k}^{K}$ with characteristics from different source domains. The [CLS] token of $z_l'$ will be used for the block-aware voting network to generate a block-aware query feature $q_l$. The block-aware voting network is simply implemented with a MLP architecture and its parameters are shared across all the MoDA blocks.

Next, we use the block query feature $q_l$ and domain prototypes $\{P_i\}_{i=1,i\neq k}^{K}$ to calculate the relevance w.r.t each source domain. We use inner product ($\langle\cdot,\cdot\rangle$) to compute similarities here for its simplicity:

$$r_l^i = \frac{1}{M_i} \sum_{m=1}^{M_i} \langle q_l, p_m \rangle \tag{8}$$

where $M_i$ is the number of identities in $i$-th source domain $D_i^s$, and $p_m$ is the $m$-th ID-specific prototypes in $D_i^s$. According to these $K-1$ relevance scores, we aggregate the intermediate features $\{h_i\}_{i=1,i\neq k}^{K}$ into one single feature. This process can be formulated as:

$$h_l^{agg} = \sum_{i\neq k}^{K-1} softmax(r_l^i) \cdot h_l^i \tag{9}$$

where $softmax(\cdot)$ is used to normalize the relevance.

And the following computation is exactly the same as the common ViT block with adapters:

$$z_l' = z_{l-1} + h_l^{agg} \tag{10a}$$

$$z_l = z_l' + MLP(LN_2(z_l')) \tag{10b}$$

In addition, we find that treating the global adapter as another domain specific adapter and incorporating it into the mixture can bring better performance. And the weight of intermediate feature of global adapter $h_l^{global}$ is fixed to 0.5, rather than being generated by the voting network.

Finally, we take the [CLS] token of the last block as the aggregated feature $F_x^{agg}$ of current sample $x$.

To optimize the block-aware voting network and MoDA blocks, the following loss functions are adopted:

(1)Image-to-text cross-entropy loss $\mathcal{L}_{i2tce}$. This part is the same as the optimization of expert adapters. We desire the aggregated feature $F_x^{agg}$ to retain inherent discriminative capability.

(2)In addition, we expect the aggregated features to be as similar as possible to those derived by domain expert and stay in close proximity in the feature space. Thus, we set two more loss functions to impose enhanced constraints: one is a consistency loss $\mathcal{L}_{consis}$, inspired by Xu et al. (2021), and the other one is a L2 loss $\mathcal{L}_{mse}$ which is naive but effective. They can be formulated as:

$$\mathcal{L}_{consis} = [\alpha_1 + \Gamma_{agg}^+ - \Gamma_{expert}^+]_+ + [\alpha_2 + \Gamma_{agg}^- - \Gamma_{expert}^-]_+ \tag{11}$$

where $\alpha_1$ and $\alpha_2$ are margins, $\Gamma_{agg}^+$ and $\Gamma_{expert}^+$ are hardest positive distances of $F^{agg}$ and $F^{expert}$ respectively, $\Gamma_{agg}^-$ and $\Gamma_{expert}^-$ are hardest negative distances of $F^{agg}$ and $F^{expert}$ respectively, $[z]_+$ equals to $max(z,0)$ Xu et al. (2021). And the L2 loss is formulated as:

$$\mathcal{L}_{mse} = \frac{1}{2}\|F^{expert} - F^{agg}\|_2^2 \tag{12}$$

The total loss can be eventually formulated as:

$$\mathcal{L}_{agg} = \mathcal{L}_{i2tce}^{agg} + \mathcal{L}_{consis} + \lambda\mathcal{L}_{mse} \tag{13}$$

$$\mathcal{L} = \mathcal{L}_{expert} + \mathcal{L}_{global} + \mathcal{L}_{agg} \tag{14}$$

Algorithm 1 in Appendix delineates the overall training procedure. During test time, all $K$ domain adapters will be used for voting network to produce the aggregated feature. Notably, we find that different approaches to mix the domain-specific feature and domain-invariant feature lead to varying performance, which will be discuss about in the ablation study.

## 4 EXPERIMENTS

### 4.1 DATASETS AND EVALUATION SETTINGS

**Datasets**. We conduct experiments on several person re-identification benchmarks following existing works Xu et al. (2021) Zhang et al. (2022): Market1501 Zheng et al. (2015), MSMT17 Wei et al.

| Method | Reference | Source | Target | | | | | | | | | | | # Tunable Params(M) |
|---|---|---|---|---|---|---|---|---|---|---|---|---|---|---|
| | | | PRID | | GRID | | VIPeR | | iLIDs | | Average | | | |
| | | | mAP | R1 | mAP | R1 | mAP | R1 | mAP | R1 | mAP | R1 | | |
| DIMN | CVPR19 | | 52.0 | 39.2 | 41.1 | 29.3 | 60.1 | 51.2 | 78.4 | 70.2 | 57.9 | 47.5 | 60* | |
| SNR | CVPR20 | M+D+C2 | 66.5 | 52.1 | 47.7 | 40.2 | 61.3 | 52.9 | 89.9 | 84.1 | 66.4 | 57.3 | 60* | |
| RaMoE | CVPR21 | +C3+CS | 67.3 | 57.7 | 54.2 | 46.8 | 64.6 | 56.6 | 90.2 | 85.0 | 69.1 | 61.5 | 63* | |
| DMG-Net | CVPR21 | | 68.4 | 60.6 | 56.6 | 51.0 | 60.4 | 53.9 | 83.9 | 79.3 | 67.3 | 61.2 | 60* | |
| QAConv$_{50}$ | ECCV20 | | 62.2 | 52.3 | 57.4 | 48.6 | 66.3 | 57.0 | 81.9 | 75.0 | 67.0 | 58.2 | 60 | |
| M$^3$L | CVPR21 | | 64.3 | 53.1 | 55.0 | 44.4 | 66.2 | 57.5 | 81.5 | 74.0 | 66.8 | 57.3 | 57 | |
| MetaBIN | CVPR21 | M+C2 | 70.8 | 61.2 | 57.9 | 50.2 | 64.3 | 55.9 | 82.7 | 74.7 | 68.9 | 60.5 | 57 | |
| ACL | ECCV22 | +C3+CS | 73.4 | 63.0 | 65.7 | 55.2 | 75.1 | 66.4 | 86.5 | 81.8 | 75.2 | 66.6 | 57* | |
| META | ECCV22 | | 71.7 | 61.9 | 60.1 | 52.4 | 68.4 | 61.5 | 83.5 | 79.2 | 70.9 | 63.8 | 165 | |
| Ours | | | **71.2** | **61.6** | **52.3** | **49.9** | **73.4** | **65.4** | **87.0** | **82.7** | **71.3** | **64.9** | **19.1** | |

Table 1: Comparison with state-of-the-art methods under protocol-1. All the images in the source domains are used for training. "*" indicates that the number of tunable parameters is estimated based on the classification head parameters under the current protocol, otherwise comes from the actual results of running code.

| Setting | Method | Reference | M+MS+CS → C3 | | M+CS+C3 →MS | | MS+CS+C3 →M | | Average | | # Tunable Params(M) |
|---|---|---|---|---|---|---|---|---|---|---|---|
| | | | mAP | R1 | mAP | R1 | mAP | R1 | mAP | R1 | |
| | SNR | CVPR20 | 8.9 | 8.9 | 6.8 | 19.9 | 34.6 | 62.7 | 16.8 | 30.5 | 50 |
| | QAConv$_{50}$ | ECCV20 | 25.4 | 24.8 | 16.4 | 45.3 | 63.1 | 83.7 | 35.0 | 51.3 | 60 |
| | M$^3$L | CVPR21 | 34.2 | 34.4 | 16.7 | 37.5 | 61.5 | 82.3 | 37.5 | 51.4 | 57 |
| Protocol-2 | MetaBIN | CVPR21 | 28.8 | 28.1 | 17.8 | 40.2 | 57.9 | 80.1 | 34.8 | 49.5 | 57 |
| | ACL | ECCV22 | 41.2 | 41.8 | 20.4 | 45.9 | 74.3 | 89.3 | 45.3 | 59.0 | 57 |
| | META | ECCV22 | 36.3 | 35.1 | 22.5 | 49.9 | 67.5 | 86.1 | 42.1 | 57.0 | 164 |
| | **Ours** | | **35.0** | **34.1** | **20.5** | **50.0** | **58.1** | **81.0** | **37.9** | **55.0** | **15.5** |

Table 2: Comparison with state-of-the-art methods under Protocol-2.

(2017), CUHK02 Li & Wang (2013), CUHK03 Li et al. (2014), CUHK-SYSU Xiao et al. (2016), PRID Hirzer et al. (2011), GRID Loy et al. (2009), VIPeR Gray & Tao (2008), and iLIDs Zheng et al. (2009). For simplicity, we denote Market1501, MSMT17, CUHK02, CUHK03, CUHK-SYSU as M, MS, C2, C3, and CS in the following. For CUHK03, we use the 'labeled' data and do not use the DukeMTMC Zheng et al. (2017) due to its privacy issues following Xu et al. (2021) Zhang et al. (2022).

**Evaluation Settings.** The mean Average Precision (mAP) and Cumulative Matching Characteristics (CMC) are used for evaluation. There are three testing protocols following the prior works Xu et al. (2021) Zhang et al. (2022) Dai et al. (2021) to evaluate the performance extensively.

For protocol-1, model is trained with both the train and test images in M+C2+C3+CS Datasets and then tested on four small datasets: PRID, GRID, VIPeR and iLIDs, respectively. Following Zhang et al. (2022) , the results are evaluated on 10 repeated random splits of query and gallery sets. The average of results will be reported. For protocol-2 and protocol-3 we choose one domain from M+MS+CS+C3 for testing and the remaining three domains for training. Protocal-2 only uses the training data of source domains, while protocal-3 uses both training and testing data of source domains for training.

## 4.2 IMPLEMENTATION DETAILS

We adopt the image encoder and text encoder of pretrained CLIP model as our backbone. For the image encoder, we use ViT-B/16, which contains 12 transformer layers with the hidden size of 768 dimensions. The parameters of both the image encoder and the text encoder are frozen. The block voting network is implemented with a FC1-GELU-LN-FC2 architecture, where FC1 layer expanding the dimensions of [CLS] token four times and the FC2 layer projects it back to the original dimension. We resize the person image size to 256 × 128. For data augmentation, we perform random cropping, random flipping, color jittering, and auto augmentation He et al. (2020) in the second stage. Similar to prior works Dai et al. (2021) Xu et al. (2021), we also discard random erasing (REA) which may effect the DG performance. We conduct all the experiments with PyTorch and the help of codebases: TransReID He et al. (2021) and FastReID He et al. (2020). All the models are trained on RTX 3080 GPU. And more details of two training stage can be found in Appendix.

| Method | | M+MS+CS $\rightarrow$ C3 | |
|---|---|---|---|
| | | mAP | Rank-1 |
| **Mixture of Expert Adapters** | w/o Global Adapter | 28.6 | 28.6 |
| | w/o Expert Adapters | 29.2 | 28.9 |
| | w/o Voting Network | 28.9 | 29.0 |
| | MoDA | 29.6 | 29.8 |
| **Mixture of Experts and Global** | w/o Global Adapter | 31.8 | 31.9 |
| | w/o Expert Adapters | 31.9 | 31.9 |
| | w/o Voting Network | 30.7 | 32.1 |
| | MoDA | **33.6** | **34.1** |

Table 3: The ablation study of effectiveness of adapters, different mixture methods and voting network on one of the protocol-2 experiments.

### 4.3 COMPARISONS TO THE STATE-OF-THE-ARTS

**Comparison under Protocol-1.** We compare our method with previous DG-ReID methods under Protocol-1, which are tested on four small datasets (PRID, GRID, VIPeR, and iLIDs). As shown in Table 1, our method could achieve competitive even better results with the SOTA methods in a more parameter-efficient way by fine-tuning much fewer parameters. And on average our method outperforms most SOTA methods.

**Comparison under Protocol-2 and Protocol-3.** We also compare our method with other methods under protocol-2 and protocol-3, as shown in Table 2 The table of results under protocol-3 can be found in the Appendix.

### 4.4 ABLATION STUDY

**The effectiveness of adapters and different mixture methods.** As shown in Table5, we find that fusing the intermediate features from the global adapter and mixture of expert adapters within each block, which produces representations containing both domain-specific and domain-invariant characteristics, can lead to considerable performance gains compared to directly concatenating the features from both in the last layer. We also show the effectiveness of the adapters in Table 5.

**The effectiveness of block voting network.** We also show the effectiveness of block-aware voting network in the Table 3. We take the summation average when w/o voting network. Here, the weight of each expert adapter is set to 1/3.

**The number of voting blocks.** Considering that the first few layers of ViT may not have extracted semantically rich features yet, we conduct ablation studies to test which layer to start voting from works best. The results show that benefiting from the block-aware design, MoDA can also help model to mix experts in a more fine-grained way by generating block-aware weight of aggregation. The results can be found in Appendix.

**The effectiveness of loss functions.** We also do the ablation study about the effectiveness of loss functions. The results can be found in Appendix.

## 5 CONCLUSION

In this work, we propose a novel MoE-based DG ReID method, named mixture of domain adapters (MoDA). We apply Adapter-tuning and CLIP to DG ReID in a parameter-efficient way. It also alleviates the linear increase of parameters with more person IDs in DG ReID models, especially MoE-based ones, by exploiting CLIP's contrastive loss with Adapters. Extensive experiments verify that MoDA achieves competitive end even better results with state-of-the-art methods with much fewer tunable parameters.

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

# A APPENDIX

## A.1 TRAINING PROCEDURE OF MODA

For the first training stage, we use the Adam optimizer with a learning rate initialized at $3.5 \times 10^{-4}$ and decayed by a cosine schedule. We train the learnable ID-specific tokens for 60 epochs and adopt the warmup strategy in the first 5 epochs. Notably, we use no augmentation in this stage. And the number of tokens $[X]_m$ is set to 4 following CLIP-ReID.

For the second training stage, the batch size is set to 64, including 16 identities and 4 images per identity. We use the Adam optimizer with a learning rate initialized at $3 \times 10^{-4}$. We train the model for 60 epochs and adopt warmip strategy in the first epochs. And the learning rate is decayed by 0.1 at the 30th and 50th epochs. Only the adapters are optimized in this stage. The weight $\lambda$ of $\mathcal{L}_{mse}$ in $\mathcal{L}_{agg}$ is set to 0.1. The margins $\alpha_1, \alpha_2$ in $\mathcal{L}_{consis}$ are both set to 0.1.

The following algorithm shows the whole training procedure of our method.

---

**Algorithm 1** Training Procedure of MoDA

---

**Require:** Training samples from source domains.
**Ensure:** A set of ID-specific tokens; Trained domain-specific adapters, global adapter and block-aware voting network.
1: **The first stage:**
2: **for** $epoch = 1$ to $max\_epoch\_stage1$ **do**
3:     Optimize tokens $[X]_m$ for each ID by $\mathcal{L}_{stage1}$.
4: **end for**
5: **The second stage:**
6: Froze all parameters except for adapters and block-aware voting network.
7: **for** $epoch = 1$ to $max\_epoch\_stage2$ **do**
8:     **for** $iter = 1$ to $max\_iter$ **do**
9:        Sample the k-th domain $D^k$ to simulate unseen target domain.
10:        **Domain-specific Adapter:**
11:        Optimize domain-specific adapter $A^k$ by $\mathcal{L}_{expert}$
12:        **Global Adapter:**
13:        Optimize global adapter $A^g$ by $\mathcal{L}_{global}$
14:        **Block-aware Voting Network:**
15:        Mixture of remaining $K - 1$ expert adapters and global adapter to produce aggregated feature $F^{agg}$
16:        Optimize block-aware voting network by $\mathcal{L}_{agg}$
17:     **end for**
18: **end for**

---

## A.2 DATASETS AND EVALUATION PROTOCOLS DETAILS

We conduct experiments on 9 public ReID mentioned in the main text of the paper. The details of these datasets are illustrated in Table 4. And in Table 5 we show the details of the three evaluation protocols adopted in our experiments.

| Datasets | # IDs | # Images | # Cameras |
|---|---|---|---|
| Market1501(M) | 1501 | 32,217 | 6 |
| MSMT17(MS) | 4101 | 126,441 | 15 |
| CUHK02(C2) | 1816 | 7,264 | 10 |
| CUHK03(C3) | 1467 | 14,096 | 2 |
| CUHK-SYSU(CS) | 11934 | 34,574 | 1 |
| PRID | 749 | 949 | 2 |
| GRID | 1025 | 1275 | 8 |
| VIPeR | 632 | 1264 | 2 |
| iLIDs | 300 | 4515 | 2 |

Table 4: The details of datasets.

| Setting | Method | Reference | M+MS+CS → C3 | | M+CS+C3 →MS | | MS+CS+C3 →M | | Average | | # Tunable Params(M) |
|---|---|---|---|---|---|---|---|---|---|---|---|
| | | | mAP | R1 | mAP | R1 | mAP | R1 | mAP | R1 | |
| Protocol-3 | SNR | CVPRJin et al. (2020) | 17.5 | 17.1 | 7.7 | 22.0 | 52.4 | 77.8 | 25.9 | 39.0 | 50 |
| | QAConv$_{50}$ | ECCVLiao & Shao (2019) | 32.9 | 33.3 | 17.6 | 46.6 | 66.5 | 85.0 | 39.0 | 55.0 | 60 |
| | M$^3$L | CVPRZhao et al. (2020) | 35.7 | 36.5 | 17.4 | 38.6 | 62.4 | 82.7 | 38.5 | 52.6 | 57 |
| | MetaBIN | CVPRChoi et al. (2020) | 43.0 | 43.1 | 18.8 | 41.2 | 67.2 | 84.5 | 43.0 | 56.3 | 57 |
| | ACL | ECCVZhang et al. (2022) | 49.4 | 50.1 | 21.7 | 47.3 | 76.8 | 90.6 | 49.3 | 62.7 | 57 |
| | META | ECCV Xu et al. (2021) | 47.1 | 46.2 | 24.4 | 52.1 | 76.5 | 90.5 | 49.3 | 62.9 | 164 |
| | Ours | | **37.7** | **38.1** | **23.7** | **51.5** | **65.0** | **84.4** | **42.1** | **58** | **15.5** |

Table 6: Comparison with state-of-the-art methods under Protocol-3.

| Protocols | Traning sets | Testing sets |
|---|---|---|
| Protocol-1 | Full-(M+C2+C3+CS) | PRID,GRID, VIPeR,iLIDs |
| Protocol-2 | M+MS+CS | C3 |
| | M+CS+C3 | MS |
| | MS+CS+C3 | M |
| Protocol-3 | Full-(M+MS+CS) | C3 |
| | Full-(M+CS+C3) | MS |
| | Full-(MS+CS+C3) | M |

Table 5: The details of evaluation protocols.

## A.3 COMPARISON UNDER PROTOCOL-3

We also compare our method with other methods under protocol-3, as shown in Table6.

## A.4 ABLATION STUDY

**The number of voting blocks.** Considering that the first few layers of ViT may not have extracted semantically rich features yet, we conduct ablation studies to test which layer to start voting from works best. The results are shown in the Table7. The results show that starting block-aware voting from the 4th layer works best. And before that the weight of each adapter is identical.

| # Voting Blocks | M+MS+CS → C3 | |
|---|---|---|
| | mAP | Rank-1 |
| 1 | 31.2 | 32.6 |
| 4 | **33.6** | **34.1** |
| 6 | 32.2 | 32.9 |
| 10 | 30.8 | 31.8 |
| 12 | 30.8 | 32.0 |

Table 7: Ablation study on the numbers of voting blocks.

**The effectiveness of loss functions.** We also do the ablation study about the effectiveness of loss functions. The results are shown in Table8. Experimental results show that applying triplet loss on the aggregated features hurts performance. The last three rows indicate that adding ID loss $\mathcal{L}_{ID}$ and classification heads, and fine-tuning the heads lead to only marginal gains but substantially more tunable parameters. And simply adding ID loss $\mathcal{L}_{ID}$ and classification heads without fine-tuning the heads leads to inferior performance.

## A.5 REFERENCES OF THE SOTA METHODS

Due to paper length limitations, the references of the SOTA methods are not cited in the original text, but are now marked in the Table6.

| Loss Functions | | | | | | | M+C3+CS-MS | | #Tunable Params |
| $\mathcal{L}_{agg}$ | | | | $\mathcal{L}_{expert}/\mathcal{L}_{global}$ | | | | | |
| $\mathcal{L}_{consis}$ | $\mathcal{L}_{mse}$ | $\mathcal{L}_{i2tce}$ | $\mathcal{L}_{triplet}$ | $\mathcal{L}_{triplet}$ | $\mathcal{L}_{i2tce}$ | $\mathcal{L}_{ID}$ | mAP | Rank-1 | |
| - | ✓ | ✓ | - | ✓ | ✓ | - | 19.3 | 48.8 | 15.5 |
| ✓ | ✓ | ✓ | ✓ | ✓ | ✓ | - | 18.8 | 45.3 | 15.5 |
| ✓ | ✓ | ✓ | - | ✓ | ✓ | ✓ | 20.6 | 50.3 | 38.1(Tune the classifier heads) |
| ✓ | ✓ | ✓ | - | ✓ | ✓ | ✓ | 20.1 | 49.2 | 15.5(Froze the classifier heads) |
| ✓ | ✓ | ✓ | - | ✓ | ✓ | - | **20.5** | **50.0** | **15.5** |

Table 8: Ablation study on the effectiveness of loss functions.

