# OpenReview forum: "MoDA: Mixture of Domain Adapters for Parameter-efficient Generalizable Person Re-Identification"
_ICLR.cc/2024/Conference — ICLR 2024 Conference Withdrawn Submission_

### Official Review · Reviewer_jvA8 · 2023-10-21

**Soundness:** 1 poor
**Presentation:** 1 poor
**Contribution:** 1 poor
**Rating:** 3
**Confidence:** 5

**Summary:**

The paper focuses on the domain generalization problem of person reID. It proposes a mixture of domain adapters to perform parameter-efficient DG reID. The authors argue that the proposed method is parameter-efficient.

**Strengths:**

None

**Weaknesses:**

1. The proposed method lacks novelty. As the author said, the proposed method is an integration of CLIP and adapter. The author just uses some hot techniques in the reid domain. I do not see any insight beyond CLIP and Adapter. I do not think this kind of paper can be accepted in any way.
2. The parameter-efficient setting may be meaningless in the DG setting; The number of tunable parameters is not a main concern in the DG reid domain. The author argue that, in the DG setting, the number of parameters of classifier heads is very large. As seen in Table 4, the sum of IDs is still very small. I have trained with 100,000 IDs, no problem exists. Also, I am not sure why you focus on this strange aspect. Fewer parameters do not mean a higher training speed and less VRAM used. I do not think the setting and the intuition of this paper is reasonable.
3. The writing is too poor; The writing of the whole paper is very poor. Also there are some typos: “blcok-aware —> block-aware”. A thorough proofreading is needed.
4. Not good experiments; in Table 6, your experimental results are much lower than others. I see the tunable parameters are small. But I do not think it is meaningful. Also, have you compared yours with the prompting methods?

**Questions:**

N/A

---

> ### Author Response · Authors · 2023-11-22
> **Reply to Reviewer jvA8**
>
> Thanks for your valuable comments firstly.
>
> **Weakness 1 & 4.** Concerning about the novelty and originality, we argue that our contribution lies not only in adopting these technologies to the Domain generalizable Person ReID domain, but also in introducing a novel block-aware voting network to exploit the text feature to utilize the cross-modality information.
>
> We believe that using both representations from text and vision feature space could strengthen the robustness of the final feature in some way. We understand the need for a detailed analysis and experimental demonstration. We will delve into this matter and provide a robust explanation in our further work.
>
> **Weakness 2.** The most significant benefit of PEFT methods comes from the reduction in memory and storage usage. We understand that this may not be a main concern in the DG ReID domain. However, it indeed helps us to scale to a larger model in downstream tasks with a much fewer tunable parameters and VRAM. And it can also alleviate the catastrophic forgetting of large models when we fine-tuning them with a downstream datasets. So, we still think it is reasonable and valuable to exploit the PEFT methods to ReID tasks including DG ReID.
>
> **Weakness 3.** We will undertake a rigorous proof-reading and ensure an immaculate grammar and improved fluency in the revised paper.
>
> Thank you again for your detailed and constructive feedback. We sincerely take your criticisms and will improve our paper in the revision.
>
> Best Regards.

---

> > ### Comment · Reviewer_jvA8 · 2023-11-22
> >
> > Many thanks for the authors' reply. It is a clear reject.

---

### Official Review · Reviewer_DXS9 · 2023-10-31

**Soundness:** 2 fair
**Presentation:** 2 fair
**Contribution:** 2 fair
**Rating:** 5
**Confidence:** 4

**Summary:**

In this work authors present a method that can handle the domain generalizable person re-identification. Inspired by the previous works in this area, authors designed a network in which they leverage the CLIP Re-ID model with a set of domain specific adapters as well as a domain generalizable adapter termed as Mixture of domain adapters (MoDA) that can efficiently scale up to larger backbones models. Another advantage of this approach is that the expert opinions are mixed even at intermediate stages so that the model can leverage finer information mixing.

**Strengths:**

The proposed approach is reasonable and clearly outlined. Despite sharing similarities with previous works, authors managed to design expert mixing strategy that results in a more parameter efficient architecture when scaling to larger backbone models for variety of source domains. Additionally, the proposed method achieves competitive results to that of state-of-the-art models when tested in equal environment under the same evaluation protocols.

**Weaknesses:**

There is no detailed discussion on the obtained results beside merely stating the method achieves better results. It is expected that some insight to be offered as why to the proposed method performs better or worse compared to certain methods. While it is important to know the method has a better accuracy, it is equally as important to know why and how that improvement occurs. Additionally, would be possible for authors to provide some failure cases and some future direction that the designed framework can be enhanced?

**Questions:**

Could authors provide some explanation on why the proposed method performs slightly better than META under protocol one but META performs better by a good margin under protocol 2?

---

> ### Author Response · Authors · 2023-11-22
> **Reply to Reviewer DXS9**
>
> Firstly, we appreciate your thoughtful analysis and the time taken to critically review our work.
>
> **Weekness 1.** Regarding your comment on the discussion of the obtained results, we understand the necessity of providing detailed insights into why and how the proposed method performs better or worse compared to certain methods. In the revised version, we plan to include a more comprehensive analysis, along with a section discussing the reasons behind the improvements.
>
> **Weekness 2 & Question 1.** We will discuss more about the analysis of our experiment results and limitation of our method. And we will try to provide some important insights and contribute to the future direction of our research.
>
> And thanks for your positive feedback on the clarity of our proposed approach and its efficiency, which really motivates us to further improve our paper. Thank you again for your time and insightful comments and look forward to further discussion.
>
> Best Regards

---

> > ### Comment · Reviewer_DXS9 · 2023-11-22
> > **Final rating**
> >
> > Based on the other reviewers comments and the provided responses by authors, I do not believe that the manuscript is ready for publication in its current form, a more critical analysis of the results and further explanation could improve the quality of the paper and provide more insight into the proposed framework. Therefore, I keep my original rating.
> >
> > Best,
> > DXS9

---

### Official Review · Reviewer_QbiD · 2023-11-02

**Soundness:** 2 fair
**Presentation:** 2 fair
**Contribution:** 2 fair
**Rating:** 3
**Confidence:** 5

**Summary:**

This study focuses on Domain Generalizable ReID (DG ReID) and introduces a new method called Mixture of Domain Adapters (MoDA) to overcome the problem that the model parameters are too large. The proposed MoDA method incorporates Global-Adapter, Export Adapter, Voting Network, and the CLIP technique for DG ReID, resulting in competitive results compared to state-of-the-art methods.

**Strengths:**

1)	The authors exploit the Global-Adapter, Export Adapter, and Voting Network for DG ReID task.

**Weaknesses:**

1)	In Tables 1 and Table 2, the bolded font is misleading, and the performance of the ACL and META methods outperforms the proposed method. In particular, the model parameter of ACL is not much larger than that of the proposed method, but the performance of ACL is much higher than that of this paper.
2)	As shown in Table 3, the Global-Adapter, Export-Adapter, and Voting Network proposed in this paper did not bring significant performance improvement, which makes me doubt the effectiveness of the method proposed in this paper.
3)	Abbreviations that appear for the first time need to indicate the full name. For example, in the third line of the abstract, the DG ReID appears for the first time without giving a full description.
4)	The GELU in Figure 2 is not given a full description, as well as in the text, which can confuse the reader.
5)	The use of mathematical symbols is not rigorous enough. The i in Formula 1 represents both the image and the index, which is easy to cause ambiguity.
6)	In Formula 3a, the definition of MA is not stated.
7)	In Formula 5, the definition of d_p and d_a are not stated.
8)	The font size in all the figures is too small.
9)	English writing needs improvement.
10)	Some grammatical errors. In the top line of section 3.2, “delineates” -> “delineate”

**Questions:**

Why does this paper not directly use images as the input to the model, but increase the optimization operation of text tokens? The implication of text tokens needs to give a more detailed analysis and experimental demonstration.

---

> ### Author Response · Authors · 2023-11-22
> **Reply to Reviewer QbiD**
>
> First of all, sincerely thanks for your time and valuable comments.
>
> **Weakness 1.** Regarding the confusion with Tables 1 and 2, it is our mistake for the potential misunderstanding arising from the bolded font. In our upcoming revision, we will clarify the numberings, and ensure a more suitable representation of the comparative performances.
>
> **Weakness 2.** Actually, we will concatenate the aggregated feature and the global feature branch feature as the final representation to perform the inference, which is inspired by META[1]. It is unclear in the Table 3 and it causes the confusion about the mixture of adapters and the concatenation of features. And then it makes the effectiveness of our proposed method and modules not as prominent. And in the revision, we will provide a clearer distinction and a more detailed explanation to strengthen our improvement.
>
> **Weakness 3-10.** Concerning the font size in our figures, typos, English grammar and writing style, we truly appreciate your patience. And we will undertake a rigorous proof-reading and ensure an immaculate grammar and improved fluency in the revised paper.
>
> **Question 1.** Actually, the model only uses images as the input both at training and inference stage. The reason why we extraly use the text tokens is that we hope to leverage a **cross-modality** approach to enhance the generalization ability of DG ReID. We believe that using both representations from text and vision feature space could strengthen the robustness of the final feature in some way.  We understand the need for a detailed analysis and experimental demonstration. We will delve into this matter and provide a robust explanation in our further work.
>
> Thank you again for your comprehensive feedback , and we believe it will lead to significant improvements in our manuscript. Grateful for your time and effort.
>
> Best Regards.
>
> [1] Boqiang Xu, Jian Liang, Lingxiao He, and Zhe Sun. Mimic embedding via adaptive aggregation: Learning generalizable person re-identification. In European Conference on Computer Vision, 2021.

---

### Official Review · Reviewer_GQT9 · 2023-11-02

**Soundness:** 3 good
**Presentation:** 2 fair
**Contribution:** 3 good
**Rating:** 5
**Confidence:** 5

**Summary:**

This paper proposes a Mixture of Domain Adapters (MoDA) framework for DG ReID, which imports CLIP and adapter to achieve the parameter efficiency during the scaling up of MOE.

**Strengths:**

Although CLIP and adapter are not new, the voting network using of the CLIP-trained ID-specific tokens to save the inference time is a reasonable and interesting idea.

**Weaknesses:**

1. The improvements in Table.1/2 is slight. The performance is even lower than SOTA. As the advantage of the proposed MoDA is based on the tunable params. I suggest the author to provide a comparison with the same level of tunable params, to see whether the performance can beat other SOTAs.
2. In Eq.11, I do not get the meaning of the Γ+agg and Γ+expert. They are two hardest positive samples, NOT two distances as described, right?
3. What is the difference between "Mixture of Expert Adapters" and "w/o Global Adapter in Mixture of Experts and Global". Moreover, in my opinion, the "Mixture of Expert Adapters" contains no global adapter, what's the meaning of "w/o Global Adapter" in "Mixture of Expert Adapters"?
4. Where is Table.5 in Section4.4?
5. Many mistakes in the writing, such as "The expter branch" in Fig.3 title.

**Questions:**

I don't like the writing of this paper, but the idea is reasonable. So I prefer to borderline.

---

> ### Author Response · Authors · 2023-11-22
> **Reply to Reviewer GQT9**
>
> Firstly, thank you for taking the time to review our work and for your insightful comments.
>
> **Weakness 1.** Regarding the performance shown in Tables 1 and 2, we acknowledge your suggestion about a comparison with the same level of tunable parameters. We believe that this recommendation will make the comparisons fairer and will emphasize parameter-efficiency advantages of our proposed method.
>
> **Weakness 2.**  Actually, Γ+agg and Γ+expert are hardest positive distances but not the specific samples. As we expect the aggregated features to be as similar as possible to those derived by domain expert, so we utilize this loss function by **minimizing the difference of the distances** between them and their respective hardest positive/negative sample.
>
> **Weakness 3.** Actually, we will concatenate the aggregated feature and the global feature branch feature as the final representation to perform the inference, which is inspired by META[1].
>
> (a) "Mixture of Expert Adapters" means that we only mix the expert adapters by the voting network in every block without global branch, and only concatenate the aggregated feature and global feature at last as we mentioned above.
>
> (b) "Mixture of Expert Adapters" means that we mix the expert adapters and global adapter by the voting network in every block and we still concatenate the aggregated feature and global feature at last.
>
> And "w/o Global Adapter" means that we only use the aggregated feature as final representation without concatenating with the global feature. And it applies to "w/o Expert Adapters" as well.
>
> It is unclear in the current version and it causes the confusion about **the mixture of adapters** and **the concatenation of features**. And in the revision, we will provide a clearer distinction and a more detailed explanation.
>
> **Weakness 4.** It seems like a typo and we actually mean the Table 3 in Section4.4. We will fix it in our revised version.
>
> **Weakness 5** **&** **Question 1.** We appreciate you pointing out typos and writing mistakes. We will thoroughly proofread the paper to avoid such errors in our revised version.
>
> Finally, thank you for your positive comment about the worthiness and strengths of our idea. We will diligently revise the paper and make it more readable, understandable, and highlight our contributions. Thank you again for your time and look forward to further discussion.
>
> [1] Boqiang Xu, Jian Liang, Lingxiao He, and Zhe Sun. Mimic embedding via adaptive aggregation: Learning generalizable person re-identification. In European Conference on Computer Vision, 2021.

---

### Official Review · Reviewer_ZKXT · 2023-11-07

**Soundness:** 2 fair
**Presentation:** 2 fair
**Contribution:** 2 fair
**Rating:** 3
**Confidence:** 4

**Summary:**

The authors propose a novel MoE-based Domain Generalizable ReID method called "Mixture of Domain Adapters" (MoDA) to address the challenges faced by existing DG ReID methods. MoDA leverages Adapter-tuning and CLIP in a parameter-efficient manner to mitigate the need for extensive fine-tuning of backbone, classifier head, and expert parameters. Their approach aims to handle the large number of person IDs typically encountered in DG ReID, allowing for better scalability to larger vision models. Many experiments demonstrate that MoDA achieves competitive or even superior results compared to state-of-the-art methods while requiring significantly fewer tunable parameters.

**Strengths:**

The authors of this paper have frozen the weights of CLIP and integrated the "Mixture of Domain Adapters" (MoDA) module for fine-tuning. They applied this technique to ReID and achieved promising experimental results. The performance seems good.

**Weaknesses:**

This paper encompasses several technologies, including CLIP, PEFT (Adapter), MoE, etc., but none of these were introduced by the authors themselves. Instead, existing technologies were adapted for use in the Re-ID domain. The authors claim, "To the best of our knowledge, our work is the first one to exploit CLIP to DG Person ReID and also the first one to attempt PEFT methods, such as adapter, for DG Person ReID." However, CLIP-ReID [1] and Clip-adapter [2] have already explored these aspects. Furthermore, AdaMix [3] and MixDA [4] have also applied MoE adapter techniques. Utilizing these technologies in the Re-ID field does not necessarily constitute a novel contribution or innovation. It is hoped that the authors can introduce more of their own design elements and emphasize improvements upon prior work.

Perhaps visualizing the relationship between MoE adapters and specific domain or person characteristics could be beneficial. For instance, in Re-ID tasks, conducting a visual analysis to understand when and under what circumstances certain adapters are activated and whether any patterns emerge could provide valuable insights.

In terms of writing, the paper reads more like a technical report combining elements MoE, CLIP, and Adapter based PEFT without a clear focus or coherence. It is suggested to start with a specific problem within DG Re-ID as the motivation, introduce the proposed solution, and emphasize the uniqueness of this approach. The method should not merely be a simple combination of existing techniques, and authors should highlight the distinctive contributions.

There is room for improvement in the style of the figures and tables as well.

[1] Li Siyuan, Li Sun, and Qingli Li. CLIP-ReID: exploiting vision-language model for image re-identification without concrete text labels. AAAI. 2023.
[2] Gao P., Geng S., Zhang R., Ma, T., Fang R., Zhang Y., Li H., and Qiao Y. Clip-adapter: Better vision-language models with feature adapters. arXiv preprint arXiv:2110.04544.
[3] Wang, Yaqing, et al. Adamix: Mixture-of-adapter for parameter-efficient tuning of large language models. arXiv preprint arXiv:2205.12410.
[4] Diao, Shizhe, et al. Mixture-of-Domain-Adapters: Decoupling and Injecting Domain Knowledge to Pre-trained Language Models Memories. arXiv preprint arXiv:2306.05406.

**Questions:**

see weakness

---

> ### Author Response · Authors · 2023-11-22
> **Reply to Reviewer ZKXT**
>
> Thanks for your insightful and valuable comments firstly.
>
> **Weakness 1.** Concerning about the novelty and originality, we argue that our contribution lies not only in adopting these technologies to the Domain generalizable Person ReID domain, but also in introducing a novel block-aware voting network to exploit the text feature to utilize the cross-modality information.
>
> Indeed, both Adamix[1] and MixDA[2] incorporate the ideas of adapters and MoE. However, in Adamix, they actually employ distinct FC-Down and FC-Up layers within one single adapter structure and primarily utilize a randomly routing mechanism. On the other hand, MixDA also does not leverage a cross-modal voting network approach. And neither of these methods specifically address domain generalization or exploitation of CLIP.
>
> We will further clarify our own design elements and improvement in the revised version of the paper more clearly.
>
> **Weakness 2.** Visualizing the relationship between MoE adapters and specific domain or person characteristics is really a valuable suggestion, which may help us to analyze our method and help readers better understand the effectiveness of our approach. We will incorporate this valuable suggestion by utilizing methods, such as T-SNE, in our revised paper and we believe insights from this analysis can enhance our contribution further.
>
> **Weekness 3.** Regarding the clarity and coherence of the paper, we appreciate your feedback. We will diligently revise the manuscript starting from the specific problem within DG Re-ID, introducing the solution we propose, and emphasizing our unique approach and its contributions.
>
> Lastly, we acknowledge that the figures and tables presentation style could be improved. We plan to polish those elements to render them more comprehensible and visually appealing.
>
> Thank you again for your constructive comments and we look forward to further discussion!
>
> Best regards.
>
>
> [1] Wang, Yaqing, et al. Adamix: Mixture-of-adapter for parameter-efficient tuning of large language models. arXiv preprint arXiv:2205.12410.
>
> [2] Diao, Shizhe, et al. Mixture-of-Domain-Adapters: Decoupling and Injecting Domain Knowledge to Pre-trained Language Models Memories. arXiv preprint arXiv:2306.05406.